# Predictive Effects of Sex, Age, Depression, and Problematic Behaviors on the Incidence and Remission of Internet Addiction in College Students: A Prospective Study

**DOI:** 10.3390/ijerph15122861

**Published:** 2018-12-14

**Authors:** Kuan-Ying Hsieh, Ray C. Hsiao, Yi-Hsin Yang, Tai-Ling Liu, Cheng-Fang Yen

**Affiliations:** 1Department of Child and Adolescent Psychiatry, Kaohsiung Municipal Kai-Syuan Psychiatric Hospital, Kaohsiung 80276, Taiwan; isanrra@gmail.com; 2Graduate Institute of Medicine, College of Medicine, Kaohsiung Medical University, Kaohsiung 80708, Taiwan; yihsya@kmu.edu.tw; 3Department of Psychiatry and Behavioral Sciences, University of Washington School of Medicine, Seattle, WA 98195, USA; rhsiao@u.washington.edu; 4Department of Psychiatry, Children’s Hospital and Regional Medical Center, Seattle, WA 98101, USA; 5School of Pharmacy, Kaohsiung Medical University, Kaohsiung 80708, Taiwan; 6Department of Psychiatry, Kaohsiung Medical University Hospital, Kaohsiung 80708, Taiwan

**Keywords:** internet addiction, predictor, incidence, remission, depression, problematic behavior

## Abstract

The aim of the study was to determine the predictive effects of sex, age, depression, and problematic behaviors on the incidence and remission of internet addiction (IA) in college students over a one-year follow-up. A total of 500 college students (262 women and 238 men) were recruited. The predictive effects of sex, age, severity of depression, self-harm/suicidal behaviors, eating problems, risk-taking behaviors, substance use, aggression, and uncontrollable sexual encounters on the incidence and remission of IA over a one-year follow-up were examined. The one-year incidence and remission rates for IA were 7.5% and 46.4%, respectively. Severity of depression, self-harm and suicidal behaviors, and uncontrollable sexual encounters at the initial investigation predicted the incidence of IA in a univariate analysis, whereas only severity of depression predicted the incidence of IA in a multivariable logistic regression (*p* = 0.015, odds ratio = 1.105, 95% confidence intervals: 1.021–1.196). A relatively young age predicted the remission of IA. Depression and young age predicted the incidence and remission, respectively, of IA in college students in the one-year follow-up.

## 1. Introduction

Internet use increased with the prevalence of mobile devices and internet-based applications. The benefits of mobile devices and applications can engender adverse effects such as internet addiction (IA). IA is defined as poorly controlled preoccupations, urges, or behaviors associated with internet use with clinically significant distress or impairment [1]. College students use the internet for studying, gaming, social networking, gambling, chatting, shopping, and watching pornographic videos. However, given that college students have free and unlimited access to the internet, have flexible schedules, and are free from their parents’ interference, they were identified as having a high risk for IA [2,3]. The prevalence of IA in college students varies among countries, ranging from 3.7% in Japan [4] to 13.6% in China [5] and 15.3% in Taiwan [6]. Identifying risk and protective factors of IA can provide critical information for developing IA prevention and intervention strategies for college students [7]. Moreover, identifying the predictors for the remission of IA can improve our understanding of the potential pathogenic mechanisms operant in IA.

IA is associated with poor physical health [8], poor academic performance [9], and interpersonal problems [10]. Moreover, IA often begins during adolescence [11]. It is reasonable to consider IA a type of problematic behavior. Problematic behaviors are a cluster of socially undesirable or prohibited behaviors, especially in teenagers [12], namely low educational achievement, conduct-disordered behavior, substance abuse, eating problems, self-injurious behaviors, and risky sexual behaviors [13]. Studies found that problematic behaviors are correlated and tend to co-occur. For example, physical aggression is strongly associated with alcohol abuse [14,15]. A comorbidity among attempting suicide, smoking, and risky sexual behaviors was also identified [16] Studies showed that IA shares common characteristics, such as maladaptive self-regulation through avoidance, relief of dysphoric symptoms, and compulsion, with substance-addictive disorders [17,18]. Alcohol use, smoking, and drug use are prevalent among individuals with IA [19,20]. Moreover, IA was found to be significantly associated with suicidality [21], aggressive behaviors [22], unprotected sexual behaviors [23], and abnormal eating behaviors [24]. However, few studies examined whether traditional problematic behaviors can predict the incidence of IA.

A study found a significant association between IA and depression [25]. In a two-year prospective study, depression was identified as a predictor of IA among women [26]. The internet may provide social support, a sense of achievement, and feeling of control and, thus, may help individuals with IA to escape from emotional difficulties [27,28]. No study examined the predictive effect of depression for incidence of IA after controlling for the effects of problematic behaviors. Studies also found that aggression is related to depression [29].

A previous study systematically reviewed both cross-sectional and longitudinal epidemiological studies of internet gaming disorder [30]. Longitudinal studies found that higher level of pathological gaming [31], diminished social competence, increased loneliness, and lower self-esteem [32], male, single-parent family, problematic video game use [33], conduct problems [34], lower levels of sport and exercise [35], positive attitude toward gaming and intention to play [36], high exploratory excitability [37], low reward dependence [37], low self-esteem [37], low family function [37], heavy episodic drinking [34], and online game playing [37] were positively related to the incidence of internet gaming disorder or IA, whereas high social integration in the classroom [33], high school-related well-being [33], high academic achievement [33], perceived behavioral control [36], and teacher autonomy support [38] were negatively related to the incidence of internet gaming disorder or IA. However, compared with the studies examining the predictors of IA incidence, the studies examining the predictors of IA remission are scarce. A previous longitudinal study found that low hostility and low interpersonal sensitivity predicted the remission of video-gaming disorder [37]. The predicting effects of age, depression, self-harm, and suicidality and uncontrollable sexual encounters on the incidence and remission of IA warrant further study.

Young age and male sex were found to be associated with IA in a cross-sectional study [39]; however, no study examined the predictive effects on the remission of IA. Although sex and age are unmodifiable, a study examining the influence of sex and age on the incidence and remission of IA can provide an understanding of the influences of sex- and age-related biological and sociocultural factors for changes in IA.

The aim of this study was to determine the predictive effects of sex, age, depression, and problematic behaviors on the incidence and remission of IA in college students over one year. We hypothesized that sex, age, depression, and problematic behaviors predict the incidence of IA, whereas sex and age predict the remission of IA in college students.

## 2. Materials and Methods 

### 2.1. Participants

Participants were recruited using online advertisements on bulletin board systems (BBS). BBS is one of main web interfaces for college students to participate in online chatting, discussion, and sharing information in Taiwan. We used online advertisements on BBS to recruit students from all universities in Taiwan. The inclusion criteria were college students aged between 20 and 30 years. Individuals who responded to the advertisement received a face-to-face interview. Those exhibiting any cognitive deficits (e.g., intellectual disability, intoxication, or dementia due to or withdrawal from substance use) that could have that prevented them from understanding the study purpose or completing the questionnaires were excluded. Informed consent was obtained prior to assessment. The study was approved by the Institutional Review Board of Kaohsiung Medical University Hospital (Ethical code: KMUH-IRB-20120249).

### 2.2. Measures

#### 2.2.1. Chen Internet Addiction Scale (CIAS)

The self-administered 26-item CIAS assesses the severity of internet-related problems in the preceding month. Each item is assessed on a four-point scale, with a total score ranging from 26 to 104 [40]. A relatively high total score indicates a more severe level of internet-related problems than a relatively low total score. The internal reliability (Cronbach’s α) of the CIAS in the present study was 0.93. Based on the result of a previous study in Taiwan [41], those with CIAS scores of 68 or higher were classified as the group with IA.

#### 2.2.2. Beck Depression Inventory-II (BDI-II) 

The self-administered 21-item BDI-II assesses the severity of cognitive-affective and somatic symptoms in the two weeks preceding the study [42]. A relatively high total BDI-II score indicates more severe depression than a relatively low total BDI-II score. The Cronbach’s α for the BDI-II in the present study was 0.88.

#### 2.2.3. Items for Assessing Behaviors on the Borderline Symptom List 

We used the 11 items on the supplement of the Borderline Symptom List (BSL-23) to assess six categories of problematic behaviors: self-harm and suicidal behaviors (three items for self-harm, suicidal plans, and suicidal attempts), eating problems (two items for binge eating and induced vomiting), risk-taking behaviors (one item for knowingly driving too fast, running around on the roofs of high buildings, and balancing on bridges), substance use (three items for getting drunk, taking drugs, and taking nonprescribed medicine or more than the prescribed dose), aggression (one item for outbreaks of uncontrolled anger or instances of physically attacking others), and uncontrollable sexual encounters (one item for uncontrollable sexual encounters of which the participants were later ashamed or that made them angry) in the week preceding the study [43]. Those who answered excepting “not at all” were all classified as having problematic behaviors. The one-month test–retest reliability (kappa) of the 11 items of BSL-23 in the present study ranged from 0.64 to 0.82.

### 2.3. Study Process and Statistical Analyses

In the initial assessment, the participants were invited to complete the CIAS, the BDI-II, and the items assessing problematic behaviors. The participants were invited to complete the CIAS one year after the initial assessment and classified into four groups based on the status of IA (Figure 1). The participants deemed not addicted to the internet in the initial assessment were stratified into groups I or II based on their subsequent not addicted or addicted status at the follow-up, respectively. The remaining participants, who were initially deemed as addicted, were stratified into groups III or IV based on remission or continuation of the addictive behavior at the follow-up, respectively. The incidence of IA and its relationships with sex, age, the severity of depression, and problematic behaviors at the initial investigation were examined in groups I and II. The remission rate of IA and its relationships with sex, age, the severity of depression, and problematic behaviors at the initial investigation were examined in groups III and IV. A chi-square test was utilized to evaluate category variables, and a *t*-test was used for continuous variables. The significant variables in the chi-square and *t*-tests were used in logistic regression analysis to examine their relationships with the incidence and remission of IA during the study period. All statistical analyses were performed using SPSS version 20.0 software (SPSS Inc., Chicago, IL, USA). A *p*-value less than 0.05 was considered statistically significant for all tests. The odds ratio (OR) and its 95% confidence intervals (CIs) were used to present significance.

## 3. Results

In total, 500 college students participated into the initial assessment. Of them, 262 were women and 238 were men. Their mean age was 22.1 years (standard deviation (SD): 1.8 years). In total, 324 participants (65.8%, 169 women and 155 men) underwent the follow-up assessment using the CIAS (Figure 1). No significant difference in the sex ratio was found between those who received and did not receive the follow-up assessment (female: 52.2% vs. 52.8%, χ^2^ = 0.021, *p* = 0.884), whereas those who did not receive the follow-up assessment were younger than those who received the follow-up assessment (21.9 vs. 22.3 years, *t* = 1.991, *p* = 0.047).

Of the 268 participants in groups I and II who did not have IA at the initial investigation, 20 were deemed as having IA in the follow-up (group II) for an incidence of 7.5% during the study period. Of the 56 individuals in groups III and IV who had IA at the initial investigation, 26 were classified as being free from IA in the follow-up (group III) for a remission rate of 46.4%.

Comparisons of sex, age, depression, and problematic behaviors between groups I and II are presented in Table 1. The results revealed that, compared with group I, group II had more severe depression (*t* = −2.862, *p* = 0.005) and were more likely to have self-harm/suicidal behaviors (χ^2^ = 4.638, *p* = 0.031) and uncontrollable sexual encounters (χ^2^ = 4.638, *p* = 0.031) at the initial investigation. There was no significant difference in IA between the college students with and without substance abuse at baseline (*p* = 0.332). At baseline, the college students with aggression were more likely to have IA than those without aggression (*p* = 0.047), whereas aggression did not predict the incidence of IA in the follow-up. Age and sex did not predict the incidence of IA. Severity of depression, self-harm and suicidal behaviors, and uncontrollable sexual encounters were used in logistic regression to analyze their relationships with the incidence of IA (Table 2). Only the severity of depression predicted the incidence of IA in the follow-up (Wals χ^2^ = 6.166, *p* = 0.015, OR = 1.105, 95% CIs: 1.021–1.196).

Comparisons of sex, age, depression, and problematic behaviors between groups III and IV are presented in Table 1. Compared with group IV, the participants in group III were younger (*t* = 2.421, *p* = 0.019), indicating that a relatively young age predicted the remission of IA. However, sex, severity of depression, and problematic behaviors did not predict the remission of IA. 

## 4. Discussion

The results of this study revealed that depression and age predicted the incidence and remission of IA, respectively, whereas problematic behaviors did not predict changes in IA in the college students during the study period. Cross-sectional studies found a significant association between depression and IA in college students [44,45]. Temperament profiles that include high harm avoidance, low self-directedness, low cooperativeness, and high self-transcendence partially account for the association between depression and IA [46]. The present study further supported the predictive role of depression for the incidence of IA. As a modifiable factor, depression should be detected early and treated to improve the mental health and prevent the incidence of IA among college students. Helping individuals with depression manage their emotional difficulties is a pertinent strategy for preventing IA [27,28].

A higher proportion of college students who developed IA in the study period had self-harm behaviors, suicidality, and uncontrollable sexual encounters at baseline than those who did not develop IA. A systematic review also found that individuals with IA are more likely to have non-suicidal self-injurious behavior and suicidality than those without IA [47]. However, the predictive effects of self-harm, suicidality, and uncontrollable sexual encounters for the incidence of IA were nonsignificant in a multivariate multiple regression analysis after the effect of depression was considered simultaneously. This result indicates that the association of self-harm, suicidality, and uncontrollable sexual encounters with the incidence of IA is mainly accounted for by depression.

The present study found that young age predicted a greater likelihood of remission of IA in the college students. A relatively young age may indicate a relatively short duration of IA, which may increase the possibility of remission of IA. Research found age differences in internet activities; for example, young age was associated with problematic online shopping [48,49]. Whether various internet activities account for young age being a predictor for the remission of IA warrants further study. 

Although research found a sex difference in IA [50,51], the present study did not support the predictive effect of sex on the incidence or remission of IA in the college students. Previous studies found that the preference of online activities differs by sex. Women tend to use social media excessively and engage in online shopping, whereas men tend to view online pornography and engage in gambling [52,53]. Further study is warranted to examine the role of sex in predicting changes in involvement in other internet activities and not only in IA. Moreover, whether sex may have various effects on the incidence and remission of IA in various age groups warrants further study.

Contrary to the hypothesis, the present study did not find significant predictive effects of eating problems, risk-taking behaviors, substance abuse, and aggression for the incidence of IA among the college students. The college students with aggression at baseline were more likely to have IA, whereas aggression did not predict the incidence of IA in the follow-up. A study found that the individuals of addiction-prone phenotypes for substance-use disorder are also sensitive to other reinforcers [54]. Research also found that alcohol use, smoking, and drug use are prevalent among individuals with IA [19,20]. Although it is reasonable to hypothesize that substance use can predict the incidence of IA, the results of the present study did not support this. Moreover, there was no significant difference in IA between the college students with and without substance abuse at baseline. Whether the association between problematic behaviors and IA exists for specific demographic or socioeconomic characteristics warrants further study.

The present study found that the remission rate of IA was 46.4% during the one-year study period. The remission rates of IA in previous studies varied because of various definitions of IA and research designs. A two-year follow-up study found that the remission rate of pathological online gaming was 16% in Dutch adolescents [32]. A one-year follow-up study found that the remission rate of online video game addiction was 50% among adolescents in Netherlands [55]. The results of the present and previous studies indicate that, like other behavioral addictions [30], IA may have the characteristic of provisionality during the period of adolescence and emerging adulthood.

Our study had several limitations. Firstly, the participants were recruited using an advertisement on BBS targeting college students. Those who did not visit BBS might not have had a chance to join this study. Secondly, the data were drawn from self-reported questionnaires, which may have resulted in shared method variance. We did not obtain side information from others to evidence participants’ levels of IA and depression and the occurrence of problematic behaviors. Thirdly, there may be factors that predict the incidence and remission of IA that were not examined in the present study. For example, the predicting effects of participants’ psychiatric diagnoses, the content of the internet activity, expectation of internet use, and peer relationship warrant further study. Finally, the rate of IA at initial assessment was 17.3%, which was comparable to the result of a previous study on college students in Taiwan [41]. However, the number of participants with remission of IA was pretty small, which may limit the statistical significance of the outcome data.

To the best of our knowledge, the present study is one of the first to examine the predictive effects of sex, age, depression, and problematic behaviors simultaneously for the incidence and remission of IA in college students. The results of the present study warrant further study to replicate. Moreover, problematic behaviors mainly occur during adolescence. The relationship between problematic behaviors and internet addiction among secondary school students warrants further study. 

## 5. Conclusions

On the basis of our study, we propose that an early survey of depression in college students is pertinent for reducing the incidence of IA. Older college students with IA are at risk of persisting IA in the subsequent year and should be the target of intervention for IA.

## Figures and Tables

**Figure 1 ijerph-15-02861-f001:**
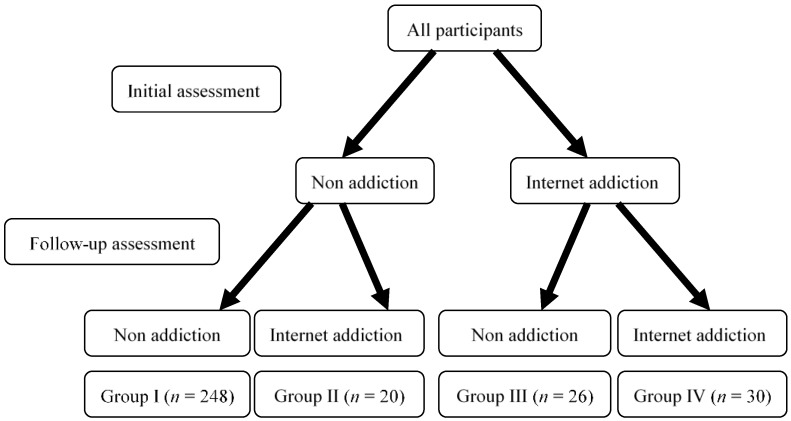
Participants in four groups with various statuses of internet addiction.

**Table 1 ijerph-15-02861-t001:** Comparisons of demographic characteristics, depression, and problematic behaviors between groups I and II and groups III and IV.

	Group I (*n* = 248)	Group II (*n* = 20)	χ^2^ or *t*	Group III (*n* = 26)	Group IV (*n* = 30)	χ^2^ or *t*
Sex, *n* (%)						
Female	129 (50.0)	11 (55)	0.066	15 (57.7)	14 (46.7)	0.678
Male	119 (48.0)	9 (45)		11 (42.3)	16 (53.3)	
Age (years), mean (SD)	22.3 (2.0)	22.4 (1.9)	−0.347	21.6 (1.2)	22.6 (1.9)	−2.421 *
Depression, mean (SD)	6.6 (5.2)	10.1 (5.3)	−2.862 **	13.3 (8.7)	12.4 (8.7)	0.382
Self-harm and suicidality, *n* (%)	5 (2.0)	2 (10)	4.638 *	3 (11.5)	2 (6.7)	0.091
Eating problems, *n* (%)	70 (28.3)	5 (25)	0.096	13 (50)	11 (36.7)	0.006
Risk taking behaviors, *n* (%)	23 (9.3)	3 (15)	0.693	5 (19.2)	3 (10)	0.299
Substance use, *n* (%)	20 (8.1)	2 (10)	0.092	5 (19.2)	6 (20)	0.363
Aggression, *n* (%)	29 (11.7)	4 (20)	1.183	8 (30.8)	7 (23.3)	0.001
Uncontrollable sexual encounter, *n* (%)	5 (2.0)	2 (10)	4.638 *	2 (7.7)	1 (3.3)	0.219

* *p* < 0.05; ** *p* < 0.01.

**Table 2 ijerph-15-02861-t002:** Predictors for incidence of internet addiction: logistic regression analysis.

	Wals χ^2^	OR	95% CIs of OR
Depression	6.166 *	1.105	1.021–1.196
Self-harm and suicidality	0.619	2.153	0.318–14.559
Uncontrollable sexual encounter	2.136	4.002	0.623–25.698

CIs: confidence intervals; OR: odds ratio; * *p* < 0.05.

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
