# Peer review of "Predictive Effects of Sex, Age, Depression, and Problematic Behaviors on the Incidence and Remission of Internet Addiction in College Students: A Prospective Study"

_ijerph, 2018, doi:10.3390/ijerph15122861_

Reviewer 1 Report

Introduction part need to be more organized. Firstly, longitudinal follow up studies regarding gaming disorders are need to cited. And, predicting factors regarding the incidence and remission are substantially different, so, the author should provide the summary of previous studies and insights regarding factor related incidence and remission, respectively.

ex. Psychiatry and Clinical Neuroscience 2017;71;425-444

Regarding the scale BSL-23, the information about reliability and validity of the measure need to be provided.

In results part, the author should provide the results about regression analysis about the factor related to remission. If the author found significant effect of age on the remission from IA just in primary comparative analysis, the author should correct the current description regarding these results and add more detailed description related to these limitation in the article. Especially, the author should provide the statistics regarding the age as a predictive factors for remission.

If the scale applied in base line assessment also applied in follow up, the results need to be described. If not, those need to be described in limitation.

I'm not sure this paper is the first article suggesting predictive effect of age, problem behaviors and depression regarding the internet addiction. The author need to be careful to write those descriptions.

Author Response

Comment

Introduction part need to be more organized.

l   Longitudinal follow up studies regarding gaming disorders are need to cited.

l   Predicting factors regarding the incidence and remission are substantially different, so, the author should provide the summary of previous studies and insights regarding factor related incidence and remission, respectively. ex. Psychiatry and Clinical Neuroscience 2017;71;425-444

Response

Thank you for your suggestions. We have revised our manuscript based on your comments.

l   Please refer to line 72-82: “A previous study systematically reviewed both cross-sectional and longitudinal epidemiological studies of Internet gaming disorder [30]. Longitudinal studies have found that higher level of pathological gaming [31], diminished social competence, increased loneliness, and lower self-esteem [32], male, single-parent family, problematic video game use [33], conduct problems [34], lower levels of sport and exercise [35], positive attitude toward gaming and intention to play [36], high exploratory excitability [37], low reward dependence [37], low self-esteem [37], low family function [37], heavy episodic drinking [34], and online game playing [37] were positively related to the incidence of Internet gaming disorder or IA, whereas high social integration in the classroom [33], high school-related well-being [33], high academic achievement [33], perceived behavioral control [36], and teacher autonomy support [38] were negatively related to the incidence of Internet gaming disorder or IA.”

l   Compare with the studies examining the predictors of IA incidence, the studies examining the predictors of IA remission are scarce. We added the results of our reviewing and added it into the revised manuscript. Please refer to line 82-87: “However, compare with the studies examining the predictors of IA incidence, the studies examining the predictors of IA remission are scarce. A previous longitudinal study found that low hostility and low interpersonal sensitivity predicted the remission of video gaming disorder [37]. The predicting effects of age, depression, self-harm and suicidality and uncontrollable sexual encounters on the incidence and remission of IA warrants further study.”

Comment

Regarding the scale BSL-23, the information about reliability and validity of the measure need to be provided.

Response

We added the test-retest reliability into the revised manuscript: “The one-month test-retest reliability (kappa) of the 11 items of BSL-23 in the present study ranges from 0.64 to 0.82.” Please refer to line 130-131.

Comment

In results part, the author should provide the results about regression analysis about the factor related to remission. If the author found significant effect of age on the remission from IA just in primary comparative analysis, the author should correct the current description regarding these results and add more detailed description related to this limitation in the article. Especially, the author should provide the statistics regarding the age as a predictive factor for remission.

Response

In the present study the variables that were significantly associated with the incidence and remission of Internet addiction in the chi-square and t tests were used in logistic regression analysis. Age was the only factor that was significantly related to the remission of Internet addiction. Therefore, age was not further selected into logistic regression analysis to examine its predicting effect on the remission of Internet addiction. We listed it as one of the limitations of this study: “There may be factors that predict the incidence and remission of IA but were not examined in the present study. For example, the predicting effects of participants’ psychiatric diagnoses, the content of the Internet activity, expectation of Internet use, and peer relationship warrant further study.” Please refer to line 240-243.

Comment

If the scale applied in base line assessment also applied in follow up, the results need to be described. If not, those need to be described in limitation.

Response

The aim of the present study was to examine the predicting factors for the incidence and remission of IA one year later. Therefore, the participants were invited to complete the CIAS 1 year after the initial assessment to determine the change of IA during the follow-up period of one year. We added the description in the revised manuscript. Please refer to line 134-135.

Comment

I'm not sure this paper is the first article suggesting predictive effect of age, problem behaviors and depression regarding the internet addiction. The authors need to be careful to write those descriptions.

Response

Thank you for your reminding. We revised this sentence into “To the best of our knowledge, the present study is one of the first to examine the predictive effects of sex, age, depression, and problematic behaviors simultaneously for the incidence and remission of IA in college students. Further study warrant to examine whether the results of the present study can be replicated.” Please refer to line 247-250.

Reviewer 2 Report

The manuscript entitled ‘Predictive Effects of Sex, Age, Depression and Problematic Behaviors on the Incidence and Remission of Internet Addiction in College Students: a Prospective Study’ presents the results of a 1-year longitudinal study on a convenience sample of 324 young adults.

The study design allows the investigation of the cause and effect of certain problem behaviors, depression, and some sociodemographic variables on the incidence and remission of internet addiction. Since the number of longitudinal studies is still low, this investigation has significant importance in understanding the predictor factors of internet addiction.

In the ‘Introduction’ part, the authors state that problematic behaviors mainly occur during adolescence. Yet, the chosen sample includes college students aged between 20 and 30. This age is not markedly adolescence, but rather emerging adulthood. It might be better to conduct research among secondary school students.

As for the ‘Method’ part, the manuscript lacks a clear description of the sampling and the participants. How did the advertising take place? Where was the study conducted (in what kind of college)? Additionally, the limitations due to the sampling method were not discussed.

Regarding ‘Discussion’, I suggest that all kinds of results of statistical analysis should be moved to the ‘Results’ part (e.g. testing the level of aggression or the amount of substance use between groups).

My additional suggestion is that the remission rate (46.4%) should be emphasized, and the possible provisionality of the problem should be pointed out.

In sum, after some minor revisions I recommend the manuscript for publishing.

Author Response

We are grateful for the valuable comments from the editors and reviewers on our manuscript. We would like to thank the reviewers for the considering our manuscript interesting and gave us many information. The following responses have been prepared to address all of the reviewers’ comments in a point-by-point fashion.

Comment

In the ‘Introduction’ part, the authors state that problematic behaviors mainly occur during adolescence. Yet, the chosen sample includes college students aged between 20 and 30. This age is not markedly adolescence, but rather emerging adulthood. It might be better to conduct research among secondary school students.

Response

Thank you for your suggestion. We agreed that the relationship between problematic behaviors and Internet addiction among secondary school students warrants further study. We added this description in line 250-252.

Comment

As for the ‘Method’ part, the manuscript lacks a clear description of the sampling and the participants. How did the advertising take place? Where was the study conducted (in what kind of college)? Additionally, the limitations due to the sampling method were not discussed.

Response

Thank you for your reminding. Participants were recruited using online advertisements on Bulletin Board Systems (BBS). BBS is one of main web interfaces for college students to online chatting, discussion and sharing information in Taiwan. We used online advertisements on BBS to recruit students from all universities in Taiwan. The inclusion criteria were college students and age between 20 and 30 years. Individuals who responded to the advertisement received a face-to-face interview. Those exhibiting any cognitive deficits (e.g., intellectual disability, intoxication, or dementia due to or withdrawal from substance use) that could have that prevented them from understanding the study purpose or completing the questionnaires were excluded. We added these descriptions in the revised manuscript. Please refer to line 99-106. Those who did not visit BBS might not have chance to join this study. Therefore, we added it as one of the limitations of the present study. Please refer to line 236-238.

Comment

Regarding ‘Discussion’, I suggest that all kinds of results of statistical analysis should be moved to the ‘Results’ part (e.g. testing the level of aggression or the amount of substance use between groups).

Response

Thank you for your suggestion. We moved the results of statistical analysis from Discussion section to Results section: “There was no significant difference in IA between the college students with and without substance abuse at baseline (p = .332). At baseline, the college students with aggression were more likely to have IA than those without aggression (p = .047), whereas aggression did not predict the incidence of IA in the follow-up.” Please refer to line 167-171.

Comment

My additional suggestion is that the remission rate (46.4%) should be emphasized, and the possible provisionality of the problem should be pointed out.

Response

Thank you for your suggestion. We added a new paragraph into the revised manuscript as below: “The present study found that the remission rate of IA was 46.4% during the one-year study period. The remission rates of IA in previous studies varied because of various definitions of IA and research designs. A two-year follow-up study found that the remission rate of pathological online gaming was 16% in Dutch adolescents [32]. A 1-year follow-up study found that the remission rate of online video game addiction was 50% among adolescents in Netherlands [55]. The results of the present and previous studies indicate that likes other behavioral addiction [30], IA may have the characteristic of provisionality during the period of adolescence and emerging adulthood.” Please refer to line 229-235.

Reviewer 3 Report

Thank you for the opportunity to review this manuscript. I found the subject matter interesting, but the type of study (correlational analysis of college students in China outside of my areas of expertise. That said, I have a number of comments for your review.

First off, studies that use convenient samples, especially college students give me pause. There is little information about the students other than their age and that they attend college. For example, were there mental health disorders in the sample obtained? And if so, what were they and how did they affect the outcomes? Furthermore, studies that rely solely on individual report are replete with measurement error and subject to bias. Most germane to this study is the likelihood that the students were accurate reporters of their behavior on internet addiction, depression and problematic behaviors (error). Moreover, how those variables (perceived or real) changed over time as a result of these study questions, maturation and numerous other variables is impossible to know. And, there is a large portion of the sample that reported not addicted at the initial assessment, and much smaller portions of the sample for the remaining three groups, questioning the statistical significance of the outcomes data.

The initial hypotheses and outcome data do however suggest that IA and depression co-occur. The directionality of the relationship is suggested and the recommendation to consider how depression may manifest in a college aged population is valuable information for international discussion, especially in the present environment where mental health services for depression and other mental health disorders are especially important considerations for policy and funding changes.

Therefore my recommendation is to accept, with a request to soften hard claims about non-statistically significant data.

Author Response

We are grateful for the valuable comments from the editors and reviewers on our manuscript. We would like to thank the reviewers for the considering our manuscript interesting and gave us many information. The following responses have been prepared to address all of the reviewers’ comments in a point-by-point fashion.

Comment

First off, studies that use convenient samples, especially college students give me pause. There is little information about the students other than their age and that they attend college. For example, were there mental health disorders in the sample obtained? And if so, what were they and how did they affect the outcomes?

Response

Thank you for your reminding. Participants were recruited using online advertisements on Bulletin Board Systems (BBS). BBS is one of main web interfaces for college students to online chatting, discussion and sharing information in Taiwan. We used online advertisements on BBS to recruit students from all universities in Taiwan. The inclusion criteria were college students and age between 20 and 30 years. Individuals who responded to the advertisement received a face-to-face interview. Those exhibiting any cognitive deficits (e.g., intellectual disability, intoxication, or dementia due to or withdrawal from substance use) that could have that prevented them from understanding the study purpose or completing the questionnaires were excluded. We added these descriptions in the revised manuscript. Please refer to line 99-106. Those who did not visit BBS might not have chance to join this study. Therefore, we added it as one of the limitations of the present study. Please refer to line 236-238. We did not determine the participants’ psychiatric diagnoses in the present study. The effect of psychiatric diagnoses on the incidence and remission of IA warrants further study. We added this description into the revise manuscript. Please refer to line 240-243.

Comment

Furthermore, studies that rely solely on individual report are replete with measurement error and subject to bias. Most germane to this study is the likelihood that the students were accurate reporters of their behaviour on internet addiction, depression and problematic behaviours. Moreover, how those variables (perceived or real) changed over time as a result of these study questions, maturation and numerous other variables is impossible to know.

Response

Thank you for your reminding. We agreed that the data were drawn from self-reported questionnaires, which may have resulted in shared method variance. We did not obtain side information from others to evidence participants’ levels of IA and depression and the occurrence of problematic behaviors. We listed it as one of limitations of the present study. Please refer to line 238-240.

Comment

And, there is a large portion of the sample that reported not addicted at the initial assessment, and much smaller portions of the sample for the remaining three groups, questioning the statistical significance of the outcomes data.

Response

Thank you for your comment. We added it as one of limitations of the present study: “The rate of IA at initial assessment was 17.3%, which was compatible to the result of the previous study on college students in Taiwan [56]. However, the number of participants with remission of IA was pretty small, which may limit the statistical significance of the outcomes data.” Please refer to line 243-246.

Round  2

Reviewer 1 Report

Thanks for your effort to revise the manuscript.